Influence of the kinesiophobia and its pain intensity relationship in subjects with onychocryptosis

Montesinos-Verdú Hipólito 1
Losa-Iglesias Marta Elena 1
Casado-Hernández Israel isracasa@ucm.es 2
Navarro-Flores Emmanuel 3
López-López Daniel 4
Cosín-Matamoros Julia 2
Pérez-Boal Eduardo 5
Muñoz-Sánchez José Luis 2
Martínez-Jiménez Eva María 2
1 Department of Nursing, Faculty of Nursing, Universidad Rey Juan Carlos , Madrid , Spain
2 Department of Nursing, Faculty of Nursing, Phisiotherapy and Podiatry, Universidad Complutense de Madrid , Madrid , Spain
3 Faculty of Nursing and Podiatry, Department of Nursing, University of Valencia, Frailty Research Organizaded Group. (FROG) , Valencia , Spain
4 Research, Health, and Podiatry Group, Department of Health Sciences, Faculty of Nursing and Podiatry, Industrial Campus of Ferrol, Universidade da Coruña , Ferrol , Spain
5 Faculty of Nursing and Phisiotherapy, University of León , León , Spain
Warnberg Julia
Electronic publication date: 2024 Sep 5
Publication date: 2024
Volume: 12
Electronic Location ID: e18022
Received 2024 Feb 16; Accepted 2024 Aug 9
Copyright: ©2024 Montesinos Verdú et al.
Copyright year: 2024
Copyright holder: Montesinos Verdú et al.
License: This is an open access article distributed under the terms of the Creative Commons Attribution License, which permits unrestricted use, distribution, reproduction and adaptation in any medium and for any purpose provided that it is properly attributed. For attribution, the original author(s), title, publication source (PeerJ) and either DOI or URL of the article must be cited.
License URL: https://creativecommons.org/licenses/by/4.0/

Keywords: Onychocryptosis, Kinesiophobia, Ingrown Nail, Pain

Funding: The authors received no funding for this work.

==============================
Background

Onychocryptosis is a nail deformity that occurs when the side of the nail grows into soft tissue, which causes pain, sepsis and the formation of granulation. The aim of the study was to evaluate and compare different levels of kinesiophobia in subjects with onychocryptosis before and after surgery to eliminate this condition.

Methods

A descriptive and observational study was conducted with a total sample size of 25 subjects with a mean age of 40.96 ± 18.25 years. The pretest sample was composed of the 25 subjects before the surgical treatment of onychocryptosis and the posttest sample was composed of the same 25 subjects after the surgical treatment of onychocryptosis. Kinesiophobia levels and total scores were self-reported using the Spanish version of the Tampa Scale for Kinesiophobia (TSK-11).

Results

The Wilcoxon test for related samples and the Mann-Whitney U test for independent samples were used to compare the results before and after the surgical treatment. It was observed that in all the items as well as in the total score, there were significant changes in the levels of kinesiophobia, after the surgical intervention for onychocryptosis (P < 0.05) compared to the levels before surgery, except for items 4 and 11 in which there were no significant differences (P > 0.05). Before surgery, 0% of the subjects with onychocryptosis reported not being afraid of movement, 16% reported mild fear of movement, 8% reported moderate fear of movement and 76% of the subjects with onychocryptosis reported severe and maximum fear of movement. On the other hand, 100% of the subjects did not report kinesiophobia after surgical treatment (P < 0.01).

Conclusions

The levels of kinesiophobia were higher in the subjects with onychocryptosis compared to the subjects after having undergone surgery to eliminate onychocryptosis.

Introduction

Approximately twenty percent of subjects who go to a family doctor with a foot problem present an ingrown toenail, also known as onychocryptosis (Geizhals & Lipner, 2019).

Onychocryptosis or ingrown nail, is a highly prevalent nail condition that occurs when the nail edge grows into the periungual dermis (Geizhals & Lipner, 2019). The great toe is most often affected and causes pain, morbidity, and impaired quality of life (Blatière, 2014; Mayeaux, Carter & Murphy, 2019). Onychocryptosis predominantly occurs in young individuals, however, congenital onychocryptosis and ingrown toenails in older individuals have been described previously. Various predisposing factors are implicated in the etiology, which include poorly fitting shoes, tight socks, hyperhidrosis, poor nail cutting technique and trauma (Thakur, Vinay & Haneke, 2020; Exley et al., 2023).

Onychocryptosis or ingrown toenail is a common condition for which there are several treatment options. Surgical matricectomy is the classic procedure, while chemical matricectomy with phenol is the most widely used today (Becerro De Bengoa Vallejo et al., 2012; Romero-Pérez, Betlloch-Mas & Encabo-Durán, 2017).

Chemical matricectomy has become an established procedure. It is performed thousands of times a year for the treatment of ingrown toenails. Many articles have described chemical matricectomy, with its many variations of technique, methods, materials, and procedures (Bostanci, Ekmekçi & Gürgey, 2001; Espensen, Nixon & Armstrong, 2002; Muriel-Sánchez et al., 2020; Vinay et al., 2022). Of these treatments, phenol chemical matricectomy appears to be the most widely used and has been associated with the fewest complications and the lowest postoperative infection rate (Giacalone, 1997; Becerro De Bengoa Vallejo et al., 2012; Vinay et al., 2022).

The term “kinesiophobia” refers to “an excessive, irrational, and debilitating fear of movement and activity resulting from a feeling of vulnerability to painful injury or re-injury”. In the long term, kinesiophobia is associated with a decrease in physical condition, avoidance of physical activity, functional disability, and depression (Domingues de Freitas et al., 2020).

The natural process of aging in people causes the toenails to thicken, making them more difficult to cut and more likely to press on the lateral skin on the sides of the nail plate, usually causing pain, becoming infected and turning into an ingrown toenail (Blatière, 2014), reducing the subject’s quality of life (Thakur, Vinay & Haneke, 2020). Some of these nail diseases have a significant impact on the subjects’ lives, both psychologically and physically (Romero-Pérez, Betlloch-Mas & Encabo-Durán, 2017). Several theories have been proposed to explain the etiology of the ingrown nail, and they can be classified in general terms according to whether the main cause is the nail itself or the soft tissues around the nail (Giacalone, 1997; Espensen, Nixon & Armstrong, 2002; Blatière, 2014; Domingues de Freitas et al., 2020).

Various psychological disorders, such as depression or lack of sociability, linked to alterations in quality of life related to general health, have been associated with musculoskeletal conditions, which may increase with greater age ranges (Roelofs et al., 2007; Gómez-Pérez, López-Martínez & Ruiz-Párraga, 2011; Heller, Manuguerra & Chow, 2016). According to previous research, there is a strong relationship between kinesiophobia and pain intensity which predispose to the future development of musculoskeletal disorders (Malfliet et al., 2017; Benatto et al., 2019; Palomo-López et al., 2020).

In addition, Becerro de Bengoa Vallejo et al. (2019) showed a better quality of life in subjects with long-standing painful ingrowing toe nails after undergoing chemical nail matricectomy surgery to treat onychocryptosis.

Several publications on classifications for cataloguing onychocryptosis along with their corresponding treatment algorithms, such as the Mozena classification system, which is based on the depth of the nail fold, the extent of infection, and granulation tissue (Mozena, 2002) or the Martínez-Nova classification, which added in the final stage hypertrophy of the distal nail edge (Martínez-Nova, Sánchez-Rodríguez & Alonso-Peña, 2007).

In addition, other onychocryptosis classifications such as the one by Heifetz and the other by Mogensen, which divides onychocryptosis into three stages according to the severity of its signs and symptoms: stage 1 of “inflammatory redness and swelling” (with mild oedema, erythema in the nail fold, and pain on pressure), stage 2 of “inflammatory discharge” (with hyperaesthesia, drainage, and infection), and stage 3 with ”granulation tissue formation”, hypertrophy of the nail fold, and worsening of the previous signs and symptoms (Heifetz, 1945; Mogensen, 1971).

The aim of the study was to assess and subsequently compare the levels of kinesiophobia in subjects with onychocryptosis before and after surgery to find the association between kinesiophobia and pain intensity caused by different degrees of onychocryptosis. In addition, the secondary objective was to predict kinesiophobia and the intensity of pain caused by this nail pathology. Therefore, we asked ourselves the following research question: Can onychocryptosis at various stages lead to kinesiophobia?.

Since no study focuses on this psychological factor in subjects with onychocryptosis, we hypothesized that people with onychocryptosis might be afraid to move.

The working hypothesis was that onychocryptosis in its different stages of classification could cause kinesiophobia, due to the fear that it may cause pain. Therefore, the purpose of the project was to determine the association between kinesiophobia and pain intensity with degrees of onychocryptosis, to quantify the degree of kinesiophobia before and after onychocryptosis treatment, to quantify pain intensity before and after onychocryptosis treatment, and to determine the degree of kinesiophobia based on the type of onychocryptosis.

Material and Methods

Design and sample

The research is a descriptive and observational study that was carried out in a private clinic between March and December 2022.

Sample size calculation

The sample size calculation was made using the difference between two groups of matched pairs with the Wilcoxon Sign test of the G*Power 3.1.9.2 software (Heinrich Heine University, Dusseldorf, Germany). A two-tailed hypothesis, an effect size of 0.70, an error probability α of 0.05, with a β level of 20% and a desired analysis power of 80% (1-β error probability) were used for sample size calculations. The final outcome of the sample size calculation was 25 subjects. A sample was recruited using consecutive sampling and a simple successive and non-randomized method. Participation selection and inclusion criteria were as follows: (1) subjects who presented onychocryptosis in any of the stages; (2) subjects of legal age; (3) healthy subjects who could walk; (4) subjects who retained sensation in the feet. The exclusion criteria for subjects were (1) refusal to give informed consent; (2) inability to understand and carry out study instructions; (3) subjects with previous surgeries for onychocryptosis.

Figure 1 Mozena onychocriptosis stages. From left to right stage I, stage IIa, stage IIb, stage III and, finally stage IV.

The different Mozena onychocriptosis stages are shown. From left to right (A) stage I, (B) stage IIa, (C) stage IIb, (D) stage III and, finally (E) stage IV.

Classification of onychocryptosis

Classification of the stages of onychocryptosis according to Mozena (2002) (Fig. 1).

Stage I: This stage is called inflammatory stage because is defined by the presence of clinical signs as erythema, mild swelling, and local pain when an external pressure is put on the medial or lateral nail fold. Usually, the nail edge does not surpass the limits of the nail plate

Stage II: This stage is called abscess stage and is made by two sub-stages. One stage called IIa, and it is characterized by clinical signs as higher pain that increase with time followed by swelling, hyperalgesia and erythema. In addition, drainage of serum and infection is characteristic. According to the nail fold, it exceeds the nail plate and the measurement of this excess is less than three mm. The other stage is the stage IIb which clinical symptoms are similar to the stage IIa, but the main difference is that a hypertrophic fold exceeds the nail plaque and it measurement is more than three mm.

Stage III: The clinical symptoms aggravate in time and are characterized by granulation tissue formation and chronic development of hypertrophy of the nail fold. In addition, hypertrophic tissue used to cover gradually the nail plate. If left untreated, the lesion will progressively worsen to chronic deformity affecting the nail and distal crease.

Stage IV: This is the final Mozena’s stages and is characterized by the final evolution of stage III. It typical clinical are excessive toenail deformity that involve the medial and lateral nail fold. The main difference with the stage III is the distal hypertrophy.

Procedure

The surgical procedure was carried out under sterile conditions in a private clinic by the same clinician with more than 10 years of experience following the Becerro De Bengoa Vallejo et al. (2012) procedure. The medial or lateral eponychium and hyponychium were separated from the nail plate using a two mm wide straight mini osteotome. A nail splitter was employed to cut the medial or lateral edge of the nail, and the nail spicule was extracted with a straight Kelly hemostat. After exposing the nail matrix, a piece of sterile gauze was completely unfolded, and one end was rolled into a pointed tip. This tip was then dipped in an 88% phenol solution and inserted through the exposed area down to the proximal matrix with the help of a two mm wide mini osteotome, ensuring that the entire exposed matrix and nail bed area was thoroughly covered. The phenol was applied to the medial side of the hallux for 4 min, followed by the lateral side for the same amount of time.

Onychocryptosis was classified into different stages during the first consultation with the subject, collecting the data related to the interpretation of the pain that the subject had at that moment, and validating it with the Spanish version of the TSK-11SV Tampa scale for Kinesiophobia. The data were collected again for each subject one month afterwards. The authors have permission to use this instrument from the copyright holders.

Each subject was managed independently by an expert clinician with more than 10 years’ experience.

The sociodemographic data included were sex (male or female), age (years), weight (kilograms), height (meters), body mass index (kilograms/centimeter squared).

Then, the subjects completed the Spanish Kinesiophobia Tampa Scale (TSK-11) (Gómez-Pérez, López-Martínez & Ruiz-Párraga, 2011). The scale consists of a psychometric detection, predictions, tracking and clinical guidance tool and was used to assess about the fear that suffer subjects when they perform a movement that involve injury areas (Roelofs et al., 2007). The TSK is a questionnaire composed by 11 items which were scored from 1 to 4. Subjects appointed their degree of agreement with each question exposed using a Likert-type scale being from 1 (totally disagree) to 4 (totally agree). The questionnaire design were used to evaluate activity grade avoidance and pain. However, regarding our study, we used the full score that was composed by a minimum parameter consist of 11 points and a maximum parameter consist of 44 points. High outcomes stated greater fear to perform a movement, which term would be, greater kinesiophobia. Additionally, TSK-11 total outcomes were defined into kinesiophobia levels as fear of perform a movement, including no fear to perform a movement (11–17 points), slight fear to perform a movement (18–24 points), moderate fear to perform a movement (25–31 points), severe fear to perform a movement (32–38 points) and maximum fear to perform a movement (39–44 points) (Cotchett et al., 2017).

In addition, psychometric features were acquired for this scale, displaying an internal consistency of 0.78 with Cronbach’s α and a test-retest with an ICC of 0.82. The repeatability and reliability of this scale in its Spanish idiom version was filled out by Gómez-Pérez, López-Martínez & Ruiz-Párraga (2011).

Ethical considerations

The study was approved by the ethics committee of the Hospital Clínico San Carlos with internal code C.I. 22/161-E (March 15, 2022).

All subjects agreed and signed the written informed consent. The recommendations of the Helsinki declaration for research involving human subjects were respected (World Medical Association, 2013).

Statistical analysis

The Shapiro–Wilks test was performed to analyze the normality distribution of the data and a normal distribution was considered if p > 0.05.

The following anthropometric characteristics were collected and detailed: age, BMI, height, stage of onychocryptosis, and weight. To compare the pretest and posttest postoperative kinesiophobia degrees of fear the Chi squared test was used. Quantitative data were studied with mean and standard deviation (SD). Due to the non-parametric data distribution of the variables, the Wilcoxon test for related samples and the Mann–Whitney U test for independent samples were used to compare the results before and after treatment.

IBM SPSS statistical software, version 19.0 (Inc, Chicago, IL, USA) was used for data analysis. A value of p < 0.05 was considered statistically significant for a confidence interval (CI) of 95%.

Results

All variables showed a non-normal distribution (P < 0.05) except for age, and body mass index (BMI) (P > 0.05) as shown in Table 1. The subjects recruited were a total of 25 subjects including, 15 women and 10 men with the characteristics shown in Table 1.

Table 1 Descriptive data of the participants’ total population by gender.

Descriptive data	Total Group Mean ± SD (95%CI) N = 25	Men Mean ± SD (95%CI) n = 15	Women Mean ± SD (95%CI) n = 10	p -value*	
Age (years)	40.96 ± 18.25 (33.42–47.49)	40.60 ± 20.63 (29.17–52.02)	41.50 ± 15.00 (30.76–52.23)	0.450*	
Weight (kg)	72.84 ± 15.17 (66.57–79.10)	80.80 ± 12.50 (73.87–87.72)	60.90 ± 10.35 (53.49–68.30)	0.000*	
Height (m)	1.69 ± 9.6 (165–1.173)	1.74 ± 8.57 (1.70–1.79)	1.62 ± 5.66 (1.58–1.66)	0.000*	
BMI (Kg/m2 )	25.03 ± 3.60 (23.54–26.52)	26.39 ± 3.16 (24.64–28.14)	22.98 ± 3.38 (20.57–25.40)	0.010*	
Onychocryptosis stage	2.2 ± 0.57 (1.96–2.43)	2.2 ± 0.70 (1.87–2.65)	2.1 ± 0.31 (1.87–2.32)	0.215*	
Notes.

BMI body mass index

Kg kilograms

M meters

SD standard deviation

CI confidence interval

* Student’s t-test for independent samples was applied. In all analyses, p < 0.05 (with a 95% confidence interval) was considered statistically significant.

Table 2 shows that in all the items as well as in the total score there are significant changes in the levels of kinesiophobia, after performing the surgical intervention for onychocryptosis (P < 0.05) compared to the levels prior to surgery, except for items 4 and 11 in which there were no significant differences (P > 0.05).

Table 2 Preoperative and postoperative kinesiophobia scores.

	Pretest (n = 25)	Posttest (n = 25)		
Item	Mean ± SD (95%CI)	Median (95%CI)	Mean ± SD (95%CI)	Median (95%CI)	p -value	
1. I am afraid of hurting myself if I exercise.	3.12 ± 1.16 (2.63–3.60)	4.00 (3.00-4.00)	1.16 ± 0.62 (0.90–1.41)	1.00 (1.00-1.00)	<0.001	
2. If I let myself be overcome by the pain, the pain would increase.	3.40 ± 1.15 (2.92–3.87)	4.00 (4.00-4.00)	1.08 ± 0.27 (0.96–1.19)	1.00 (1.00-1.00)	<0.001	
3. My body is telling me that I have something serious.	3.16 ± 0.98 (2.75–3.56)	3.00 (3.00-4.00)	1.00 ± 0.00 (0.00–0.00)	1.00 (1.00-1.00)	<0.001	
4. Having pain always means that there is an injury in the body.	3.36 ± 0.70 (3.07–3.64)	3.00 (3.00-4.00)	2.92 ± 0.86 (2.56–3.27)	3.00 (3.00-3.00)	0.063	
5. I am afraid of accidentally injuring myself.	2.48 ± 1.22 (1.97–2.98)	2.00 (2.00-3.85)	1.20 ± 0.57 (0.96–1.43)	1.00 (1.00-1.00)	<0.001	
6. The safest way to avoid increasing pain is to be careful and not make unnecessary movements.	3.16 ± 1.14 (2.68–3.63)	4.00 (3.00-4.00)	1.08 ± 2.27 (0.96–1.19)	1.00 (1.00-1.00)	<0.001	
7. It wouldn’t hurt so much if I didn’t have something serious in my body.	3.28 ± 0.84 (2.93–3.62)	3.00 (3.00-4.00)	1.08 ± 0.27 (0.96–1.19)	1.00 (1.00-1.00)	<0.001	
8. The pain tells me when to stop the activity so as not to injure myself.	3.52 ± 0.82 (3.18–3.85)	4.00 (3.14-4.00)	1.44 ± 0.71 (1.14–1.73)	1.00 (1.00-1.85)	<0.001	
9. It is not safe for a person with my disease to do physical activities.	3.00 ± 1.11 (2.53–3.46)	3.00 (2.00-4.00)	1.16 ± 0.62 (0.90–1.41)	1.00 (1.00-1.00)	<0.001	
10. I can’t do everything normal people do because I could easily get injured.	2.60 ± 1.19 (2.10–3.09)	3.00 (2.00-3.85)	1.04 ± 0.20 (0.95–1.12)	1.00 (1.00-1.00)	<0.001	
11. No one should be physically active when they are in pain.	3.40 ± 0.95 (3.00–3.79)	4.00 (3.00-4.00)	2.88 ± 1.16 (2.39–3.36)	3.00 (2.14-4.00)	0.079	
TSK-11 Total score	34.48 ± 7.04 (31.57–37.38)	35.00 (33.00-38.00)	16.04 ± 2.50 (15.00–17.07)	16.00 (15.00-16.00)	<0.001	
Notes.

SD standard deviation

CI confidence interval

IR interquartile range

TSK-11 Tampa Scale for Kinesiophobia

P value from U Mann Whitney. In all analyses, p < 0.05 (with a 95% confidence interval) was considered statistically significant.

Table 3 shows that 0% of onychocryptosis subjects reported no fear of movement, 16% report slight fear of movement, 8% reported moderate of fear of movement and 76% reported severe and maximum fear of movement. On the other hand, 100% of the subjects did not report kinesiophobia after surgical treatment (P < 0.01).

Table 3 Kinesiophobia with different degrees of fear of movement pre and postoperative.

Outcome measurements	Pretest (n = 25)	Posttests (n = 144)	p -value (cases vs controls)	
TSK-11 Category*	Fear of movement	No	0 (0%)	25 (100%)	<0.01	
		Slight	4 (16%)	0 (0%)		
		Moderate	2 (8%)	0 (0%)		
		Severe	10 (40%)	0 (0%)		
Maximum	9 (36%)	0 (0%)	
Notes.

Abbreviations: * TSK-11, Tampa Scale for Kinesiophobia. Frequency, percentage (%) and Chi-squared test (χ2) were utilized. TSK-11 categories were divided as follows: (1) 11 to 17 points: no fear of movement, (2) 18 to 24 points: slight fear of movement, (3) 25 to 31 points: moderate fear of movement, (4) 32 to 38 points: severe fear of movement, (5) 39 to 44 points: maximum fear of movement. †, TSK-11 scores, Median ± interquartile range, range (min–max) and the Kruskal–Wallis test were used. In all the analyses, p < 0.05 (with a 95% confidence interval) was considered statistically significant.

Discussion

Onychocryptosis is a nail disorder that causes losing quality of life (Blatière, 2014). We have not found any study in the scientific literature that links fear of movement with onychocryptosis. For this reason, we believe that this study is important. We have tried to demonstrate the relationship between kinesiophobia and the different degrees of onychocryptosis, since our hypothesis was that subjects with onychocryptosis would have kinesiophobia.

According to the sociodemographic outcomes we found statistically significant differences in weight, height and BMI by gender. BMI male values were 26.39 ± 3.16 kg/cm2, on the other hand the BMI female values were 22.98 ± 3.38 kg/cm2. The BMI male mean values showed overweight was one prevalent risk factor according to Arica et al. (2019a), who investigated the clinical and sociodemographic characteristics of 207 subjects with ingrowing toenails and displayed that the obesity percentage was 34.1% from the total sample size. These findings were similar in our research (Arica et al., 2019a).

Previous research by Sayilan et al. (2022) about kinesiophobia, functional level, mobility, and pain in older adults after surgery did not showed statistically significant differences by gender in preoperative and postoperative pain in a sample size consisted of 61 women and 38 men. In contrast, the research performed by Soetanto, Chung & Wong (2006) in the Chinese population to investigate is there was differences by gender in pain perception concluded that female population showed more pain at the pain tolerance level and male population reported the association between anxiety outcomes with a higher pain.

Our research was performed in adults with a 40,96 years +- 18.25 mean age. Previous research by Arica et al. (2019a), Arica et al. (2019b) showed that an ingrown nail is a clinical condition that can occur at any age, but it is more prevalent in early adulthood. This condition is highly uncomfortable and painful for the patient, often resulting in workforce losses. The increased incidence of ingrown toenails in people aged 30–40 years is believed to be linked to an active lifestyle, including work and sports activities, and for women, factors such as pregnancy and higher body mass indices. While ingrown toenails are commonly found in all pediatric age groups, they are particularly frequent in adolescents. Among adolescents, the most prevalent type is 1, highlighting the significance of improper nail cutting. In adults, out of 718 observed cases of ingrown toenails, 41.4% were classified as stage 1, 44.5% as stage 2, and 14.1% as stage 3 (Arica et al., 2019a; Arica et al., 2019b). According to our outcomes, the stage 2 of onychocryptosis was the most prevalent in the study, consistent with the results of Arica. This stage is usually the most painful due to its clinical progression, which is why it tends to have a greater painful impact in adulthood.

Therefore, we used the scale (TSK-11) to compare the levels of kinesiophobia in 25 subjects before and after surgery to eliminate onychocryptosis.

In all the items, as in the total score, there were significant changes in the levels of kinesiophobia, after surgery for onychocryptosis (P < 0.05) compared to the levels prior to surgery, except for items 4 and 11 in that there were no significant differences (P > 0.05).

Our results showed that before surgery, 0% of the subjects with onychocryptosis reported not being afraid of movement, 16% reported mild fear of movement, 8% reported moderate fear of movement, 40% reported severe fear of movement and 16% reported maximum fear of movement. On the other hand, 100% of the subjects did not report kinesiophobia after surgical treatment (P < 0.01).

We believe that the fear of movement is caused by the pain subjects have, since we found many other studies where kinesiophobia is related to musculoskeletal pain such as hallux valgus pain, fibromyalgia or migraine (Malfliet et al., 2017; Benatto et al., 2019); (Palomo-López et al., 2020).

There are many studies on kinesiophobia, most of them at the musculoskeletal level, on back pain, at the lumbar level , or at the spinal cord level (Comachio et al., 2018; Luque-Suarez, Martinez-Calderon & Falla, 2019; Domingues de Freitas et al., 2020; Van Bogaert et al., 2021), total knee arthroplasty, heel pain and, after upper extremity injuries; there are currently no studies associating kinesiophobia and pain intensity with degrees of onychocryptosis (Cotchett et al., 2017; Filardo et al., 2017; Bartlett & Farnsworth, 2021).

Regarding future lines of research, we believe that it would be interesting to evaluate the appearance of exostosis in the distal phalanx of the hallux since in many cases it could be the main cause of pain in subjects with onychocryptosis as well as obesity, venus pressure, unsuitable shoes and bad nail cutting.

However, our study has limitations, since the sample size was small, and it would be beneficial to have a larger sample size. In addition, it would be beneficial to be able to use complementary exploration material such as X-rays or ultrasound to rule out other underlying pathologies such as subungual exostoses or periungual corns.

Conclusion

Subjects with onychocryptosis develop kinesiophobia which causes losing quality of life. Levels of kinesiophobia were significantly higher in the subjects before surgery compared to after surgery. In addition, it was observed that the highest levels of kinesiophobia occurred in the most advanced stages of onychocryptosis; therefore, if we can control the pain, we could reduce kinesiophobia.

Supplemental Information

Data S1 Raw data

Supplemental Information 2 Strobe checklist

Additional Information and Declarations

Competing Interests

Author Contributions

Human Ethics

Data Availability

The authors declare there are no competing interests.

Hipólito Montesinos-Verdú conceived and designed the experiments, performed the experiments, prepared figures and/or tables, authored or reviewed drafts of the article, and approved the final draft.

Marta Elena Losa-Iglesias conceived and designed the experiments, performed the experiments, analyzed the data, authored or reviewed drafts of the article, and approved the final draft.

Israel Casado-Hernández conceived and designed the experiments, performed the experiments, prepared figures and/or tables, authored or reviewed drafts of the article, and approved the final draft.

Emmanuel Navarro-Flores conceived and designed the experiments, analyzed the data, authored or reviewed drafts of the article, and approved the final draft.

Daniel López-López conceived and designed the experiments, analyzed the data, authored or reviewed drafts of the article, and approved the final draft.

Julia Cosín-Matamoros conceived and designed the experiments, performed the experiments, authored or reviewed drafts of the article, and approved the final draft.

Eduardo Pérez-Boal conceived and designed the experiments, authored or reviewed drafts of the article, and approved the final draft.

José Luis Muñoz-Sánchez conceived and designed the experiments, performed the experiments, authored or reviewed drafts of the article, and approved the final draft.

Eva María Martínez-Jiménez conceived and designed the experiments, authored or reviewed drafts of the article, and approved the final draft.

The following information was supplied relating to ethical approvals (i.e., approving body and any reference numbers):

The study was approved by the ethics committee of the Hospital Clínico San Carlos

The following information was supplied regarding data availability:

The raw data are available in the Supplemental File.

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
