# Peer review of "Influence of the kinesiophobia and its pain intensity relationship in subjects with onychocryptosis"

_PeerJ, doi:10.7717/peerj.18022_

## Round 0.1 · original submission · Major Revisions

Thank you for submitting your manuscript to our journal. After three reviews, we have identified a few areas where additional information would significantly enhance the clarity and comprehensiveness of your study.

Please address all reviewers' comments. Additionally, the sample size is small for such a common condition. Expanding on the rationale for this sample size, or increasing the sample size if feasible, would strengthen the validity of your results.

Kind regards,

Reviewer 1 ·

Basic reporting

I've read the paper "Pain Intensity Evaluation in a Population with Ingrown Toenail: A Descriptive Study" with great interest. The paper aims to evaluate and compare different levels of kinesiophobia in subjects with onychocryptosis before and after surgery aimed at eliminating this condition.

However, I have some significant concerns regarding the paper. Firstly, the title does not accurately reflect the main focus of the study, as the primary variable measured is kinesiophobia rather than "pain," as indicated in the title.

Additionally, there appear to be some inaccuracies in the introduction. The stages of onychocryptosis described do not align with the authors' references.

Experimental design

A sample of 25 patient in this very frequent condition too short to take in account the results.

One of the glaring omissions I've noticed is that the paper aims to compare kinesiophobia in subjects with onychocryptosis before and after surgery, yet there is no explanation provided regarding the surgical procedure itself.

Were all patients operated on by the same surgeon?
What surgical technique was employed?
Were there any reported complications?
How much time elapsed between the surgery and the subsequent kinesiophobia test?

The absence of answers to these questions within the paper renders it challenging to fully comprehend the study and its findings.

Validity of the findings

The validity of the findings is questionable due to several factors, including the small sample size, methodological shortcomings, and the inherent likelihood that a painful condition would improve following surgery.

·

Basic reporting

I don't see the added value of mentioning the link between psychological disorders and quality of life, general health and musculoskeletal conditions (scentences 59-63)
No further comments

Experimental design

reference of G power doesn't mention (Heinrich Heine University, Dusseldorf, Germany)
Altough it's not 100% relevant it would be interesting to add some details on surgical procedures which were used.
No further comments

Validity of the findings

No comments

Additional comments

/

·

Basic reporting

0.- Abstract
Abstract is well structured in different sections and include the aim of the study. Results and conclusions are clearly explained.

1.- Introduction.
Line 35 : Is “bad cutting technique” also not one of the most common causes ? Or is this not mentioned in literature ?
Lines 37-38. Authors should add reference about this sentence.
Lines 41-43. Authors refer that “many articles have described …” but only one is referenced. Please add more references.
Lines 53 and 54. There are two numbers that are supposed to be references. Please correct this issue.
Lines 67 to 69 seems to be the same in lines 73 to 77. Please correct this mistake. Please, rewrite.
Line 70 : please rephrase to a more academic level.

2.- Material and methods
The material and methods section is very well explained.
In statistical analysis section authors must include the TSK 11 outcomes using the chi squared test comparing the pretest and posttest postoperative kinesiophobia degrees of fear that is showed in table 3.


3.- Results
Are clearly explained. Concise and impactful . No remarks.

4.- Discussion
Did authors find any difference by gender in suffering in growing toenail?
Do male vs. female experience the same or have they different pain levels ? Please explain and add reference.
On the other hand, the sample mean age is 40,96 years +- 18.25. Is the ‘injury’ the same in younger / elderly people ? Can age influence pain levels ?

5.- Conclusion
Are well explained based on the aim of the study.

Experimental design

No comment

Validity of the findings

No comment

Additional comments

Critical Appraisal of the Manuscript Titled :

"Pain Intensity Evaluation in Population with Ingrown Toenail: A Descriptive Study"

In general :
The manuscript under review offers a comprehensive exploration of pain intensity assessment within a cohort afflicted by ingrown toenails. It investigates the intricate interplay between ingrown toenails and their well-known impact on individuals' quality of life, particularly focusing on the repercussions/advantages of surgical interventions.
This is a new type of study, in my opinion, which has never been published before and therefor very promising.

Structural Evaluation:
The manuscript demonstrates commendable structural integrity. However, minor refinements are warranted to enhance its cohesion and clarity further. (see remarks below)

Scientific Rigor and Methodological Precision:
The study exhibits a commendable adherence to scientific rigor, evidenced by its methodological framework. Nevertheless, elucidation on the precise methodologies employed for pain intensity evaluation, alongside delineation of participant selection criteria and motivation for e.g. sample size determination, would augment methodological transparency.

Insights into Pain Intensity Dynamics:
The findings offer compelling insights into the multifaceted nature of pain experienced by individuals grappling with ingrown toenails. However, a more nuanced explanation of pain intensity metrics (and statistical analyses) could be recommended to provide a comprehensive understanding of the study's outcomes.

Contextualization and Discussion:
Well done !
Clear.
Few suggestions :
While the discussion provides a cogent synthesis of the study's findings, further contextualization within the broader landscape of ingrown toenail pathology and therapeutic modalities would fortify its scholarly impact even more !!!
Furthermore, clarification on the limitations inherent in the study design and recommendations for future research, not only looking for possible appearance of exostosis, could be very interesting.

Language Precision and Terminological Accuracy:
No remarks.

Conclusive Remarks:
In summary, the manuscript presents a noteworthy contribution to the scientific discourse surrounding ingrown toenails and their attendant pain dynamics. With some small revisions and adjustments, to address the outlined areas of improvement, it has promise for dissemination within scholarly circles and clinical domains alike.

---

## Round 0.2 · accepted · Accept

Thank you for submitting your manuscript to PeerJ. After thorough evaluation by four independent reviewers and careful consideration of the feedback provided, I am pleased to inform you that your manuscript has been accepted for publication. I have no further comments or suggestions.

I look forward to seeing your work published!

Congratulations on your successful submission.

Reviewer 4 ·

Basic reporting

I thank the authors for their comments. I have read through the subsequent changes made to the manuscript and I have no further comments or suggestions. I hope that this publication is available to the public shortly.

Experimental design

no comments

Validity of the findings

no comments